# Recent Advances of Indium Oxide-Based Catalysts for CO_2_ Hydrogenation to Methanol: Experimental and Theoretical

**DOI:** 10.3390/ma16072803

**Published:** 2023-03-31

**Authors:** Dongren Cai, Yanmei Cai, Kok Bing Tan, Guowu Zhan

**Affiliations:** Integrated Nanocatalysts Institute (INCI), College of Chemical Engineering, Huaqiao University, 668 Jimei Avenue, Xiamen 361021, China

**Keywords:** CO_2_ hydrogenation, methanol, indium oxide-based catalysts, oxygen vacancies, hydrogen dissociation

## Abstract

Methanol synthesis from the hydrogenation of carbon dioxide (CO_2_) with green H_2_ has been proven as a promising method for CO_2_ utilization. Among the various catalysts, indium oxide (In_2_O_3_)-based catalysts received tremendous research interest due to the excellent methanol selectivity with appreciable CO_2_ conversion. Herein, the recent experimental and theoretical studies on In_2_O_3_-based catalysts for thermochemical CO_2_ hydrogenation to methanol were systematically reviewed. It can be found that a variety of steps, such as the synthesis method and pretreatment conditions, were taken to promote the formation of oxygen vacancies on the In_2_O_3_ surface, which can inhibit side reactions to ensure the highly selective conversion of CO_2_ into methanol. The catalytic mechanism involving the formate pathway or carboxyl pathway over In_2_O_3_ was comprehensively explored by kinetic studies, in situ and ex situ characterizations, and density functional theory calculations, mostly demonstrating that the formate pathway was extremely significant for methanol production. Additionally, based on the cognition of the In_2_O_3_ active site and the reaction path of CO_2_ hydrogenation over In_2_O_3_, strategies were adopted to improve the catalytic performance, including (i) metal doping to enhance the adsorption and dissociation of hydrogen, improve the ability of hydrogen spillover, and form a special metal-In_2_O_3_ interface, and (ii) hybrid with other metal oxides to improve the dispersion of In_2_O_3_, enhance CO_2_ adsorption capacity, and stabilize the key intermediates. Lastly, some suggestions in future research were proposed to enhance the catalytic activity of In_2_O_3_-based catalysts for methanol production. The present review is helpful for researchers to have an explicit version of the research status of In_2_O_3_-based catalysts for CO_2_ hydrogenation to methanol and the design direction of next-generation catalysts.

## 1. Introductions

The greenhouse effect caused by excessive CO_2_ emission has seriously threatened the survival of human beings and other organisms [1,2,3,4]. In order to cope with the current grim situation, many countries have established a target timeline to reach the peak of CO_2_ emission and achieve carbon neutrality. For example, China has promised to realize a carbon peak by 2030 and carbon neutrality by 2060 [5,6,7,8]. Therefore, CO_2_ capture and utilization (CCU) technology has attracted much attention [9,10,11,12,13]. In particular, the use of “green hydrogen” produced with renewable energy to convert waste CO_2_ into methanol is not only able to effectively reduce CO_2_ emission but also can store renewable energy in liquid fuel, which is an important method to realize resource utilization of CO_2_ [14,15,16,17,18,19].

CO_2_ hydrogenation to methanol mainly includes CO_2_ to methanol reaction (1), reverse water gas shift reaction (RWGS, 2), and CO to methanol reaction (3), respectively [20,21,22,23]. Reactions (1) and (3) are exothermic; thus, low temperature is conducive to the formation of methanol but hinders the activation of CO_2_. On the contrary, competitive reaction (2) is an endothermic reaction, which is significantly promoted at high temperatures, resulting in a sharp decrease in methanol selectivity [24,25,26]. Therefore, the development of efficient catalysts with the aim to decrease CO_2_ activation energy and promote methanol formation at a suitable temperature is the key to realizing the industrial application of CO_2_ hydrogenation to methanol.
CO_2_ + 3H_2_ ⇆ CH_3_OH + H_2_O ∆H_298 K_ = −49.5 kJ/mol(1)
CO_2_ + H_2_ ⇆ CO + H_2_O ∆H_298 K_ = 41.2 kJ/mol(2)
CO + 2H_2_ ⇆ CH_3_OH ∆H_298 K_ = −90.7 kJ/mol(3)

Currently, several types of materials have been used as the catalysts for CO_2_ hydrogenation to methanol, including Cu [15,27,28,29], noble metals (Pt, Pd, Ru, etc.) [30,31,32,33], M_a_ZrO_x_ (M_a_ = Ga, Zn, etc.) solid solution [34,35,36,37], and indium oxide (In_2_O_3_) [21,38,39,40]. Among them, Cu-based catalysts have been widely investigated in CO_2_ hydrogenation to methanol [41]. However, they are prone to the Ostwald ripening effect and particle migration under high temperatures and water surroundings, which results in catalyst deactivation [42,43,44]. By comparison, noble metals exhibit high stability and resistance to sintering and poisoning, so they are regarded as an alternative to Cu-based catalysts. Nevertheless, these catalysts are not able to efficiently catalyze the reaction and regulate the product distribution due to the weak binding with CO_2_ molecules [45,46]. Our previous research also confirmed this conclusion, which reveals that when Pt catalyzes CO_2_ hydrogenation alone, no methanol generates, and the selectivity of CO is as high as 100% [5,47]. Moreover, M_a_ZrO_x_ solid solution, particularly ZnZrO_x_, is a potential catalyst for CO_2_ hydrogenation to methanol [35,48,49,50]. In ZnZrO_x_, Zn is doped into ZrO_2_ lattice by replacing Zr (Zn-Zr-O_x_), and the solid solution structure provides reaction sites of Zn and adjacent Zr for activating H_2_ and CO_2_, respectively (synergistic effect), thus producing methanol with high selectivity [51]. However, the low activity and the mobility of ZnO still limit the applications of the catalysts [36].

To our knowledge, In_2_O_3_ has been regarded as a highly selective catalyst for CO_2_ hydrogenation to methanol in recent years [52]. It is generally believed that CO_2_ can be adsorbed and activated by oxygen vacancies on the In_2_O_3_ surface, which are periodically generated and annihilated to inhibit the occurrence of side reactions, therefore hydrogenating CO_2_ to methanol with high selectivity [53,54,55]. Not only does In_2_O_3_ show higher methanol selectivity than Cu and noble metals, but it also exhibits higher catalytic activity than ZnO [21]. Additionally, In_2_O_3_ can be further supported and modified to promote the activation of CO_2_ and H_2_, and stabilize the key reaction intermediates, thus presenting great potential to become an excellent catalyst for the sustainable and efficient production of methanol. Herein, we gave a comprehensive overview of the recent advances of In_2_O_3_-based catalysts for CO_2_ hydrogenation to methanol. The active site and mechanism of pure In_2_O_3_ catalyst for CO_2_ hydrogenation were stated at first. Then, the discussion was concentrated on two important strategies, namely metal doping and hybrid with other metal oxides, to enhance the catalytic activity of In_2_O_3_ by promoting the dissociation of hydrogen to hydrogenate intermediates and the formation of oxygen vacancies to activate CO_2_ and stabilize the key intermediates. Some suggestions in the future study were finally proposed to improve the performance of In_2_O_3_-based catalysts for CO_2_ hydrogenation to methanol from the experimental and theoretical aspects. This review focused on the regulation and modification of active sites of In_2_O_3_-based catalysts to facilitate the activation of reactants and stabilization of intermediates in CO_2_ hydrogenation, which is conducive to the design of more efficient In_2_O_3_-based catalysts in future studies.

## 2. In_2_O_3_-Based Catalysts for CO_2_ Hydrogenation to Methanol

### 2.1. Pure In_2_O_3_ Catalyst

The idea of In_2_O_3_ as the catalyst for CO_2_ hydrogenation to methanol stems from its excellent CO_2_ selectivity in methanol steam reforming (MSR) reactions [56,57]. Based on density functional theory (DFT), Ge et al. [53] predicted the feasibility of CO_2_ hydrogenation to methanol catalyzed by In_2_O_3_ (110) with oxygen vacancies. They proposed that In_2_O_3_ would inhibit RWGS reaction, and methanol was the major product on the surface of defective In_2_O_3_ (110). As shown in Figure 1, the reaction process obeys the mechanism of periodic generation and annihilation of oxygen vacancies, including adsorption and activation of CO_2_ on oxygen vacancies, CO_2_ hydrogenation to form intermediate species, methanol desorption, and regeneration of oxygen vacancies. To confirm the research results of DFT, Liu et al. [38] used commercial In_2_O_3_ activated at a high temperature (500 °C) as the catalyst for CO_2_ hydrogenation. The experimental results demonstrated that methanol yield increased with the increase in reaction pressure; however, due to the limitation of thermodynamics, it increased first and then decreased as the temperature increased. In addition, 2.82% of methanol yield and 3.69 mol h^−1^ kg_cat_^−1^ of methanol production rate were obtained at 330 °C and 4 MPa, which was superior to many other catalysts. In 2016, Pérez-Ramírez et al. [58] revealed that nano In_2_O_3_ can efficiently catalyze CO_2_ hydrogenation to methanol, obtaining more than 0.18 g_MeOH_ h^−1^ g_cat_^−1^ of space-time yield. They also found that compared to Cu/ZnO/Al_2_O_3_, the highest methanol yield of In_2_O_3_ was achieved at 300 °C, indicating that In_2_O_3_ can maintain high methanol selectivity at higher temperatures. Two years later, they reported the mechanism and microkinetics of methanol synthesis from CO_2_ hydrogenation over In_2_O_3_ [59]. The results indicated that the apparent activation energy experimentally determined for CO_2_ hydrogenation to methanol (103 kJ mol^−1^) was lower than that of the RWGS reaction (117 kJ mol^−1^), which explains the superior methanol selectivity over In_2_O_3_. In_2_O_3_ (111) was experimentally and theoretically proved to be the most exposed surface termination, indicating CO_2_ can be activated by oxygen vacancies surrounded by three indium atoms. In addition, the most favorable pathway to methanol comprises three consecutive additions of hydrides and protons, which features CH_2_OOH* and CH_2_(OH)_2_* as intermediates. In 2019, by an operando examination, Müller et al. [60] proved that In_2_O_3−x_ was the active phase of methanol synthesis, while In^0^ led to the deactivation of the catalyst.

Liu et al. [61] prepared an In_2_O_3_ catalyst by precipitation method for CO_2_ hydrogenation to methanol. The results showed that under the operating conditions (H_2_/CO_2_ molar ratio of 4, the volume space velocity of 21,000 cm^3^ h^−1^ g_cat_^−1^, reaction pressure of 5 MPa, and reaction temperature of 300 °C), the CO_2_ conversion and methanol space-time yield were 9.4% and 0.335 g_MeOH_ h^−1^ g_cat_^−1^. Guo et al. [62] investigated the catalytic activity of cubic bixbyite-type indium oxide (*c*-In_2_O_3_) and rhombohedral corundum-type indium oxide (*r*-In_2_O_3_) in CO_2_ hydrogenation to methanol. Due to the impressive reducibility and reactivity, *c*-In_2_O_3_ was higher than *r*-In_2_O_3_ in CO_2_ conversion; however, *r*-In_2_O_3_ possessed higher methanol selectivity because of weaker methanol and stronger CO adsorption. Moreover, the in situ DRIFTS experiments revealed that CO_2_ could be reduced to CO via redox cycling and hydrogenated to methanol via the formate pathway. In addition, Sun et al. [63] successfully designed an In_2_O_3_ nanocatalyst with higher catalytic activity under the guidance of theoretical calculation, which suggested that the hexagonal In_2_O_3_ (104) surface had a far superior catalytic performance. As shown in Figure 2, the experimental results also confirmed that compared to cubic In_2_O_3_ (*c*-In_2_O_3_), a novel hexagonal In_2_O_3_ (*h*-In_2_O_3_-R) with a high proportion of the exposed (104) surface exhibited higher catalytic activity and possessed high stability. Moreover, Li et al. [54] investigated the dissociative adsorption of H_2_ during CO_2_ hydrogenation over cubic and hexagonal In_2_O_3_ by DFT, and they found that the oppositely charged In and O pair sites on the reduced In_2_O_3_ surfaces played a significant role in facilitating the heterolytic dissociation of H_2_, which contributed to the formation of anionic hydride around the In sites to promote CO_2_ hydrogenation to methanol. Additionally, *h*-In_2_O_3_ (104) surface is considered the best surface for CO_2_ hydrogenation to methanol due to the facile formation of the oxygen vacancies at low coverage and the favorable formation of the hydride adsorbate at the In sites.

In the last two years, the research topic of CO_2_ hydrogenation to methanol over In_2_O_3_ has still attracted considerable interest. Based on the solvothermal method, Wu et al. [64] successfully fabricated mixed-phase indium oxide with controllable cubic and hexagonal phases to enhance catalytic performance in CO_2_ hydrogenation to methanol. Due to its enhanced textural properties and oxygen vacancies, mixed-phase *c*/*h*-In_2_O_3_ catalysts demonstrated higher CO_2_ conversion and space-time yield of methanol and kept stable in the reaction. To understand the structure–activity relationship, Nørskov et al. [65] systematically studied the methanol synthesis over In_2_O_3_ (111) and In_2_O_3_ (110) by combining DFT calculations with microkinetic modeling. The theoretical activity volcano shown in Figure 3 suggested that catalytic activity was closely related to the number of reduced In layers on In_2_O_3_ surfaces, specifically, for In_2_O_3_ (110), a surface oxygen vacancies between 0.17 and 1 ML (ML: the top layer, from surface to interior) possessed the highest catalytic activity, while for In_2_O_3_ (111), the number of oxygen vacancies should be increased to 1~5 ML to obtain the optimal activity. Similarly, Gao et al. [66] revealed the structure–performance relationship of cubic In_2_O_3_ catalyst in CO_2_ hydrogenation via the study of reaction mechanism and catalytic activities at all the different surface oxygen vacancy sites on stable (111) flat surface, (110) flat surface, and (110) step surface. The conclusion was that the rate-determining step of methanol synthesis for a given oxygen vacancy site can be determined by the stability of H_2_COO* and CH_2_O* intermediates along with the formation energy of the oxygen vacancy sites, and tri-coordinated oxygen vacancy sites were beneficial to the formation of methanol, whereas bi-coordinated oxygen vacancy sites favor CO formation. CO_2_ hydrogenation to methanol on indium-terminated In_2_O_3_ (100), defective In_2_O_3_ (110), and In_2_O_3_ (111) surfaces were also deeply investigated by Zhang et al. [67]. It was found that the adsorbed CO_2_ was preferable to form HCOO* compared with CO* and COOH* and underwent HCOO*, H_2_CO*, and H_3_CO* intermediates due to the lowest energy barriers. The defective In_2_O_3_ (110) was proven to be the optimal surface for CO_2_ hydrogenation to methanol, while the indium-terminated In_2_O_3_ (100) surface displayed the lowest catalytic activity. In addition, Creaser et al. [39] proposed a kinetic model based on Langmuir–Hinshelwood–Hougen–Watson (LHHW) mechanism for CO_2_ hydrogenation to methanol over In_2_O_3_ catalyst. The model revealed that RWGS was obviously enhanced at high temperatures, causing methanol synthesis to reverse (methanol steam reforming, MSR). Apparent activation energies for CO_2_ hydrogenation to methanol and RWGS were 90 and 110 kJ mol^−1^, respectively, over In_2_O_3_ derived from the experimental data. The results obtained from these detailed investigations were conducive to the development of reliable reactor and process designs.

Although In_2_O_3_ exhibited excellent methanol selectivity in CO_2_ hydrogenation, the low CO_2_ conversion limited the methanol yield. Therefore, based on the cognition of the In_2_O_3_ active site and the reaction pathway of CO_2_ hydrogenation over In_2_O_3_, two strategies shown in Figure 4 were adopted to enhance the performance of In_2_O_3_, including (I) introducing other metal elements into In_2_O_3_ and (II) combining In_2_O_3_ with other metal oxides. The catalytic performance of In_2_O_3_-based catalysts is summarized in Table 1.

### 2.2. Metal/In_2_O_3_ Composite Catalysts

The abundant oxygen vacancies in In_2_O_3_ can adsorb and activate CO_2_, and the periodic generation and annihilation of oxygen vacancies can inhibit the side reactions, leading to the highly selective conversion of CO_2_ to methanol. However, the weak hydrogen adsorption and dissociation of In_2_O_3_ limit the hydrogenation of carbon species, so CO_2_ conversion is very low. Accordingly, the introduction of a noble metal or transition metal (M) could improve CO_2_ conversion due to the synergistic catalysis of M and In_2_O_3_. As shown in Figure 5, the H_2_ molecule was adsorbed and activated on the M surface to generate H active species (step ①) and then combined with lattice oxygen of In_2_O_3_ via spillover (step ②) to create the oxygen vacancies (step ③). CO_2_ molecule was adsorbed and activated by the obtained oxygen vacancies (step ④) and finally hydrogenated to methanol by combining with H active species (step ⑤).

#### 2.2.1. Noble Metal/In_2_O_3_ Catalysts

**Pd/In_2_O_3_ catalyst.** Many investigations have been concentrated on Pd/In_2_O_3_ catalyst for CO_2_ hydrogenation to methanol in recent years. Ge et al. [87] studied methanol synthesis from CO_2_ hydrogenation over Pd/In_2_O_3_ by the DFT method. They found that the HCOO* route competes with the RWGS route over Pd/In_2_O_3_ in the reaction process, and H_2_COO* + H* ⇆ H_2_CO* + OH* and cis-COOH* + H* ⇆ CO* + H_2_O* were their rate-limiting steps, respectively. The HCOO* route was the major pathway for methanol synthesis from CO_2_ hydrogenation. Moreover, the H adatom activated by the Pd cluster and H_2_O on the In_2_O_3_ substrate was extremely significant for the promotion of methanol production, and the adsorbed hydroxyl on the interface of Pd/In_2_O_3_ can induce the transformation of the Pd_4_ cluster, which caused the change in final hydrogenation step. According to the guidance of the theoretical study, they prepared Pd/In_2_O_3_ with high dispersion of Pd nanoparticles by thermal treatment of Pd-peptide composite/In_2_O_3_ for methanol synthesis from CO_2_ hydrogenation [68]. The prepared catalyst exhibited much more excellent activity than that of pure In_2_O_3_ due to the better ability to adsorb and dissociate H_2_ for hydrogenation steps and the formation of oxygen vacancies. As a result, such a catalyst was able to demonstrate 20% of CO_2_ conversion, 70% of methanol selectivity, and 0.89 g_MeOH_ h^−1^ g_cat_^−1^ of space-time yield (STY), respectively.

Huang et al. [69] from our group detailly investigated the effect of strong metal–support interaction between Pd and In_2_O_3_ on the catalytic performance of CO_2_ hydrogenation to methanol by adjusting the morphology of In_2_O_3_. The results indicated that the combination of Pd and hollow In_2_O_3_ nanotubes derived from MIL-68(In) nanorod was more conducive to the methanol production compared with other morphologies of In_2_O_3_, which was due to more formation of Pd^2+^ via electron transfer from Pd to the curved In_2_O_3_ (222) to enhance H_2_ adsorption and formation of surface oxygen vacancies. In addition, to prevent the formation of the In-Pd bimetallic phase that led to the quick deactivation of the catalyst, our group further developed TCPP(Pd)@MIL-68(In) as precursors to prepare Pd/In_2_O_3_ [88]. Compared to PdCl_2_, TCPP(Pd) (metalloporphyrins) can be served as a capping agent for the growth of MIL-68(In) and a shuttle for transporting the Pd^2+^, thereby improving the dispersion of Pd during the process of calcination and reduction, and preventing excessive reduction to form In-Pd bimetallic phase. Both theoretical and experimental results indicated that the prepared Pd/In_2_O_3_ possessed excellent thermodynamic selectivity for methanol. For the same purpose of reducing the formation of In-Pd alloy, Zhan et al. [89] from our group adopted rape pollen pretreated by hydrochloric acid as the biological template to fabricate hierarchically structured bio-In_2_O_3_ and bio-In_2_O_3_/Pd, as shown in Figure 6. The results suggested that the pollen template with acid etching possessed a hollow cage-like structure and abundant functional groups (viz., -COOH and -NH_2_) on the surface, which was conducive to the growth of In_2_O_3_ with abundant superficial oxygen vacancies. Compared to the sample without acid pretreatment (bio-In_2_O_3_-0/Pd), bio-In_2_O_3_-15/Pd demonstrated a better ability to inhibit the formation of In-Pd alloy due to the more uniform In_2_O_3_ spatial distribution to reduce the interaction between Pd and In_2_O_3_. In the following research, our group further developed bifunctional catalyst Pd/In_2_O_3_/H-ZSM-5 for dimethyl ether synthesis from CO_2_ hydrogenation, whereby Pd/In_2_O_3_ prepared by carbonized alginate templating favored CO_2_ hydrogenation into methanol, and H-ZSM-5 favored methanol dehydration into dimethyl ether. Compared to commercial Pd/In_2_O_3_ (Com-PdIn), microspherical-confined nano In_2_O_3_ possessed more excellent texture properties to disperse the Pd nanoparticles, thus obtaining more than 450 g_MeOH_ kg_cat_^−1^ h^−1^ of STY, whereas Com-PdIn only achieved 50.8450 g_MeOH_ kg_cat_^−1^ h^−1^ of STY [90].

Pérez-Ramírez et al. [91] reported an effective coprecipitation method to incorporate isolated palladium atoms into an In_2_O_3_ lattice for forming low-nuclearity palladium clusters, which can overcome the selectivity and stability limitations associated with palladium nanoparticles. Additionally, to disperse the active components highly, Zhang et al. [92] employed the citric acid method to load In_2_O_3_ and Pd on SBA-15, respectively. It can be found that oxygen vacancies were promoted with increasing Pd amount. The as-prepared catalyst possessed excellent performance with 12.9% of CO_2_ conversion, 83.9% of methanol selectivity, and 1.1 × 10^−2^ mol_MeOH_ h^−1^ g_cat_^−1^ of STY, which was due to the high dispersion of In_2_O_3_ and Pd nanoparticles on SBA-15, and the synergetic effect of H_2_ dissociation on Pd species and CO_2_ activation on In_2_O_3_. Moreover, Wu et al. [70] introduced Mn and Pd into In_2_O_3_ to improve the methanol selectivity and CO_2_ conversion. The results showed that Pd species were highly dispersed on the MnO/In_2_O_3_ due to the strong metal–support interactions, and 1 wt% Pd/MnO/In_2_O_3_ exhibited excellent activity (240.6 g_MeOH_ kg_cat_^−1^ h^−1^ of STY) and stability in CO_2_ hydrogenation.

**Pt/In_2_O_3_ catalyst.** The combination of Pt and In_2_O_3_ for CO_2_ hydrogenation to methanol has also been reported. For instance, Li et al. [71] adopted the coprecipitation method to synthesize Pt/In_2_O_3_ and investigated the effect of Pt content on the catalytic performance. They found that as Pt content increased, CO_2_ conversion increased, whereas methanol selectivity increased first and then decreased. The highly dispersed Pt^n+^ was embedded into the In_2_O_3_ lattice to promote the formation of oxygen vacancies and contribute to CO_2_ activation. In the reaction process, the unstable Pt^n+^ was reduced to Pt nanoparticle, and the stable Pt^n+^ kept the high dispersion. Both Pt^n+^ and Pt can activate H_2_, but the effect on the reaction was quite different; specifically, the highly dispersed Pt^n+^ was used as the Lewis acid site to promote H_2_ dissociation for CO_2_ hydrogenation to methanol, while Pt nanoparticles induced the RWGS reaction to decrease the methanol selectivity. Similarly, Liu et al. [61] supported Pt on In_2_O_3_ to improve the methanol yield. The results showed that the CO_2_ conversion and methanol yield over Pt/In_2_O_3_ were 17.3% and 0.542 g_MeOH_ h^−1^ g_cat_^−1^ at 300 °C, respectively (In_2_O_3_: 9.4% and 0.335 g_MeOH_ h^−1^ g_cat_^−1^). As compared to In_2_O_3_, Pt/In_2_O_3_ possessed more excellent catalytic stability, which was mainly due to the high dispersion of Pt nanoparticles and strong interaction between Pt and In_2_O_3_ to inhibit the excessive reduction in In_2_O_3_. In addition, to keep the high dispersion of Pt, Pan et al. [93] synthesized Pt/film/In_2_O_3_ catalyst shown in Figure 7 via the cold-plasma/peptide-assembly (CPPA) method. The prepared Pt/film/In_2_O_3_ obtained 37.0% of CO_2_ conversion and 62.6% of methanol selectivity at 30 °C and 0.1 MPa in a dielectric barrier discharge (DBD) plasma reactor. The film of the catalyst played significant roles in the improvement of catalytic performance, namely inhibiting the agglomeration of Pt nanoparticles and transferring the electrons from the catalyst to CO_2_. The results of this work provided a valuable reference for CO_2_ hydrogenation to methanol at room temperature and pressure. Pérez-Ramírez et al. [94] highlighted flame spray pyrolysis as a synthesis platform to assess metal (Pt, Ni, Au, etc.) promotion in In_2_O_3_-based catalysts for CO_2_ hydrogenation. Compared to Ni clusters or Au nanoparticles, the atomically dispersed and well-stabilized Pt had a more obvious promoting effect on In_2_O_3_ for CO_2_ hydrogenation to methanol. Moreover, DFT simulations further revealed that the high concentration of isolated Pt atoms could greatly enhance homolytic H_2_ splitting and increase the availability of hydrides for C-H hydrogenation due to the formation In_3_Pt and In_2_Pt_2_ ensembles, therefore facilitating methanol production.

**Other noble metal/In_2_O_3_ catalyst.** In addition to Pd and Pt, other noble metals were also introduced into In_2_O_3_ to promote catalytic performance. Shrotri et al. [73] found that methanol STY over In_2_O_3_-based catalyst can be improved from 0.18 g_MeOH_ h^−1^ g_cat_^−1^ to 1.0 g_MeOH_ h^−1^ g_cat_^−1^ after doping of Rh. This was because, on the one hand, Rh promoted the dissociation of H_2_ to lead to the formation of more oxygen vacancies on the In_2_O_3_ surface. On the other hand, Rh was related to the production of formate species with a low activation barrier confirmed by DFT. Similarly, Liu et al. [72] also investigated the influence of Rh addition to In_2_O_3_ on methanol production from CO_2_ hydrogenation. They demonstrated that the existence of Rh can enhance the dissociative adsorption and spillover of hydrogen, which was instrumental in surface oxygen vacancies formation of In_2_O_3_ and CO_2_ activation, so the STY of 0.5448 g_MeOH_ h^−1^ g_cat_^−1^ over Rh/In_2_O_3_ was obtained while it was only 0.3402 g_MeOH_ h^−1^ g_cat_^−1^ over In_2_O_3_. In addition, they also supported Ru [74], Au [75], Ir [76], and Ag [95] on the In_2_O_3_ for CO_2_ hydrogenation to methanol, and the results indicated that the catalytic activity could be enhanced to a great extent.

#### 2.2.2. Base Metal/In_2_O_3_ Catalysts

**Ni/In_2_O_3_ catalysts.** Recently, Ni/In_2_O_3_ catalysts have also attracted wide attention in methanol production from CO_2_ hydrogenation. In 2020, Liu et al. [77] prepared an In_2_O_3_-supported nickel catalyst (Ni/In_2_O_3_) by a wet chemical reduction for CO_2_ hydrogenation, and the results suggested that the highly dispersed Ni species can be used as active sites for hydrogen dissociation and spillover to contribute to the formation of oxygen vacancies and hydrogenation process. Therefore, the effective synergy of Ni sites and In_2_O_3_ support resulted in superior catalytic performance, specifically, 18.47% of CO_2_ conversion, more than 54% of methanol selectivity, and 0.55 g_MeOH_ h^−1^ g_cat_^−1^ of STY at 300 °C and 5 MPa. Subsequently, to further understand the superior catalytic performance of Ni/In_2_O_3_, they investigated the synergistic effect of the metal–support interaction and interfacial oxygen vacancies on methanol synthesis via DFT calculation [96]. It was found that the interfacial oxygen vacancies were beneficial for boosting the CO_2_ adsorption and charge transfer between the nickel species and indium oxide, synergistically promoting the selectivity of methanol. Simultaneously, among the three reaction pathways examined (formate pathway, CO hydrogenation, and RWGS pathway, respectively), the RWGS pathway was proven to be the most theoretically favored for methanol synthesis from CO_2_ hydrogenation over Ni/In_2_O_3_, as shown in Figure 8. In addition to the above research work, they also introduced ZrO_2_ into Ni/In_2_O_3_ catalyst (Ni/In_2_O_3_-ZrO_2_) for CO_2_ hydrogenation to methanol [97]. The solid solution formed by ZrO_2_ and In_2_O_3_ can optimize and stabilize the oxygen vacancies of In_2_O_3_ to avoid the excessive reduction in the bulk indium oxide, thus possessing a 43.2% increase in STY of methanol. Different from the traditional synthesis method, Hensen et al. [78] combined Ni with In_2_O_3_ using flame spray pyrolysis (FSP) synthesis. The obtained NiO-In_2_O_3_ catalyst possesses high specific surface areas and block morphology. When NiO loading is 6 wt%, ~0.25 g_MeOH_ h^−1^ g_cat_^−1^ of STY can be obtained over the corresponding catalyst at the conditions of 250 °C and 30 bar. The comprehensive characterizations revealed the strong interactions between Ni cations and In_2_O_3_ when NiO loading is lower 6 wt%, which contributed to the promotion of surface density of oxygen vacancies. Additionally, DFT calculation suggested that the introduction of Ni species lowered the energy barrier of H_2_ dissociation to facilitate hydrogenation of adsorbed CO_2_ on oxygen vacancies.

**Other metal/In_2_O_3_ catalysts.** To improve the performance of In_2_O_3_, Qi et al. [79] prepared In_x_-Co_y_ oxides catalysts for CO_2_ hydrogenation to methanol. It was found that the methanation activity catalyzed by Co species was suppressed, and the best catalyst (In_1_-Co_4_) exhibited nearly five times methanol STY compared to that of pure In_2_O_3_ at conditions of 300 °C and 4 MPa. Several in situ and ex situ characterizations suggested that CO_2_ hydrogenation over Co species and In_x_-Co_y_ oxides all followed the formate pathway, and much stronger adsorbed capacity of CO_2_ and carbon-containing intermediates on In_x_-Co_y_ oxides catalyst contributed to a feasible surface C/H ratio, therefore facilitating CH_3_O* to produce methanol instead of being over-hydrogenated to methane. Gascon et al. [80] explored metal–organic framework (MOF) mediated synthetic approaches to prepare a Co_3_O_4_-supported In_2_O_3_ catalyst for CO_2_ hydrogenation to methanol. Compared to the traditionally coprecipitated In@Co catalytic system, the induction period in the hydrogenation process over MOF-derived In@Co catalyst could be tuned because ZIF-67(Co) support provided better In dopant distribution. In addition, the sequential pyrolysis-calcination steps could promote the formation of mixed-metal carbide (Co_3_InC_0.75_) to stabilize high In distribution and prevent the formation of large individual oxide domains, thus leading to a faster induction period. The prepared catalyst (used 3In@8Co(300)) showed nanoparticles featuring core–shell morphologies (Co-In oxides shell over Co_3_InC_0.75_ core) shown in Figure 9 and could obtain 0.65 g_MeOH_ h^−1^ g_cat_^−1^ of maximum STY with methanol selectivity of 87% at conditions of 250 °C and 50 bar. Additionally, based on ZIF-67(Co), Zhang et al. [98] obtained a Co/C-N catalyst through the pyrolysis method and then mixed it with In_2_O_3_ in different methods to prepare In_2_O_3_/Co/C-N for CO_2_ hydrogenation to methanol. It was found that the proximity of Co/C-N and In_2_O_3_ played a significant role in the synergetic catalysis for methanol synthesis from CO_2_ hydrogenation. Moreover, the obvious difference in placement of separate Co/C-N and In_2_O_3_ in catalytic performance also indicated CO_2_ might be adsorbed and activated on the surface of In_2_O_3_ to form carbon intermediates and then were further hydrogenated into methanol or byproducts over Co/C-N surface. Furthermore, the existence of the N element could improve the electron interaction of Co and In_2_O_3_ and prevent the sintering of In_2_O_3_ particles, thereby increasing the catalytic activity and stability for CO_2_ hydrogenation to methanol.

Additionally, the combination of Cu and In_2_O_3_ also can be a good choice to improve the catalytic performance. Wu et al. [81] employed the coprecipitation method to fabricate various CuO-In_2_O_3_ and investigated the effect of the Cu:In molar ratio on the physicochemical properties and catalytic activity for methanol synthesis. The prepared catalyst mainly exhibited in the form of Cu_11_In_9_ phase and In_2_O_3_ at low Cu:In molar ratio (≤1:2) after reduction treatment or in the reaction process, whereas with the increase in Cu content, Cu_7_In_3_ phase was continuously weakened, and Cu phase emerged, which resulted in the formation of Cu-Cu_7_In_3_-In_2_O_3_. CuIn(1:2) catalyst obtained maximum methanol STY (5.95 mmol_MeOH_ h^−1^ g^−1^) at the conditions of 260 °C and 3.0 MPa due to the highest Cu dispersion and the highest surface oxygen vacancies concentration, and the synergistic effect, Cu_7_In_3_ phase for H_2_ dissociation and In_2_O_3_ for CO_2_ adsorption, were considered as the major contributions for the efficient catalytic efficiency. The interfacial sites between Cu and metal oxides (In, Zn, and Zr) were tuned by Yu et al. for CO_2_ hydrogenation to methanol [99]. The results suggested that the introduction of In_2_O_3_ into Cu/ZrO_2_ catalyst can increase the methanol formation rate from 52.7 mmol g_cat_^−1^ to 60.5 mmol g_cat_^−1^. This was because, on the one hand, the formation of Cu_x_In_y_ surface species inhibited the RWGS reaction on the Cu surface. On the other hand, ZrO_2_ stabilized the In_2_O_3_ and generated additional In-Zr mixed oxide sites for CO_2_ conversion to methanol.

### 2.3. In_2_O_3_/Metal Oxides Composite Catalysts

Combining In_2_O_3_ with other metal oxides is also a significant strategy, which can improve the dispersion of In_2_O_3_, increase the content of oxygen vacancies for CO_2_ adsorption, and stabilize the key intermediates to facilitate methanol formation from CO_2_ hydrogenation. Supporting In_2_O_3_ on ZrO_2_ is the most common and effective method because the electronic structure effect and crystal lattice mismatching between In_2_O_3_ and ZrO_2_ are beneficial to CO_2_ activation for the formation of methanol. The research results by Pérez-Ramírez et al. proved that combining In_2_O_3_ with ZrO_2_ can enhance the catalytic activity and stability of CO_2_ hydrogenation to methanol [58]. On the one hand, the reduced Zr centers can attract oxygen atoms from the active phase in the reaction process, therefore increasing oxygen vacancies for CO_2_ adsorption and activation. On the other hand, ZrO_2_ support effectively improved the dispersion of In_2_O_3_ nanoparticles. Next, they explored the electronic, geometric, and interfacial phenomena between In_2_O_3_ and ZrO_2_ [86]. The results suggested that the catalytic performance of mixed In-Zr oxides could not be improved by coprecipitation, thereby excluding the primary role of electronic parameters. The epitaxial growth of In_2_O_3_ was permitted on both monoclinic and tetragonal ZrO_2_; however, the more obvious lattice mismatching contributes to the lower dispersion of In_2_O_3_ on monoclinic ZrO_2_. Detailed characterizations and kinetic analyses revealed two major facilitation of monoclinic ZrO_2_ support for In_2_O_3_ performance. One is that the epitaxial alignment of In_2_O_3_ on monoclinic ZrO_2_ ensured the high dispersion of the oxide to prevent sintering. The other is that the less favorable lattice matching between In_2_O_3_ and monoclinic ZrO_2_ produces tensile strain more easily, favoring the formation of oxygen vacancies on In_2_O_3_. The strong electronic oxide–support interaction between In_2_O_3_ and ZrO_2_ for CO_2_ hydrogenation to methanol was investigated by Gong et al. through quasi-in situ XPS experiments and DFT calculation [82]. Compared to the combination of In_2_O_3_ and tetragonal ZrO_2_ (In_2_O_3_/t-ZrO_2_), In_2_O_3_/m-ZrO_2_ (m-: monoclinic) exhibits more excellent catalytic performance (CO_2_ conversion up to 12.1% with methanol selectivity of 84.6%) due to the stronger interaction to lead to the high dispersion of In-O-In over m-ZrO_2_. Methanol synthesis from CO_2_ hydrogenation over In_2_O_3_/m-ZrO_2_ follows the formate pathway. It was confirmed that the electron was transferred from m-ZrO_2_ to In_2_O_3_ to generate electron-rich In_2_O_3_, which can facilitate the dissociation of H_2_ and help HCOO* transform into CH_3_O* by hydrogenation. Blum et al. [100] paid important attention to the support effect and surface reconstruction of In_2_O_3_/m-ZrO_2_ in the process of CO_2_ hydrogenation to methanol. They proposed that the modifying effects of m-ZrO_2_ on In_2_O_3_ mainly had two aspects: (I) m-ZrO_2_ serves as a reservoir for partially reduced In_2_O_3_ (InO_x_, 0 < x <1.5) due to the fact that InO_x_ can semireversibly migrate in and out of the subsurface of m-ZrO_2_ under reaction conditions (623 K). The decrease in surface InO_x_ concentration at high temperatures resulted in the low selectivity toward methanol and a rapid increase in RWGS reaction. (II) The interaction that Zr centers attracted the O atom of In_2_O_3_ led to the activation of the In-O bond at the In_2_O_3_-m-ZrO_2_ interface to generate oxygen vacancies, and the high dispersion of In_2_O_3_ nanoparticles on m-ZrO_2_ prevented the over-reduction of In_2_O_3_ under catalytic conditions compared to the bare In_2_O_3_. Based on their work, they also summarized the reaction mechanism pathway on the bare In_2_O_3_ and In_2_O_3_/m-ZrO_2_, as exhibited in Figure 10. Witoon et al. [101] studied the effect of the calcination temperature of ZrO_2_ support on the physicochemical properties and catalytic activities of In_2_O_3_/ZrO_2_ for converting CO_2_ and H_2_ into methanol at a high reaction temperature. As the calcination temperature increased (from 600 to 1000 °C), the crystal of ZrO_2_ support gradually changed from an amorphous phase to a tetragonal phase. The high calcination temperature of ZrO_2_ support can decrease the reduction degree of In_2_O_3_, indicating the better interaction between In_2_O_3_ and tetragonal ZrO_2_ compared to amorphous ZrO_2_. In addition, the adsorption capacity of prepared In_2_O_3_/ZrO_2_ catalysts for CO_2_ and H_2_ was enhanced with the increase in calcination temperature of ZrO_2_ support, which promoted the highly selective conversion of CO_2_ and H_2_ into methanol instead of methane, whereas it did not have a significant impact on the formation of CO.

Müller et al. [102] investigated the effect of the ZrO_2_ phase on the reducibility, local structure, and catalytic performance of In_2_O_3_/ZrO_2_ for CO_2_ hydrogenation to methanol by operando X-ray absorption spectroscopy (XAS) and XRD studies. The results suggested that the amorphous ZrO_2_ (am-ZrO_2_) support could not form a solid solution with In_2_O_3_, and led to the rapid reduction in In_2_O_3_ to pure In^0^ under reaction conditions, therefore suffering deactivation within minutes. For tetragonal ZrO_2_ (t-ZrO_2_) support, although it can inhibit the complete reduction of In_2_O_3_ into In^0^, the reduction extent was still too great (an average oxidation state of In below +2), resulting in poor catalytic activity. Surprisingly, it was found that the interaction between In_2_O_3_ nanoparticles and monoclinic ZrO_2_ (m-ZrO_2_) can impel atomical dispersion of In^2+^/In^3+^ into m-ZrO_2_ lattice to form solid solution m-ZrO_2_:In, which prevented the over-reduction of In species (an average oxidation state of +2.3) and stabilized the active In-oxygen vacancy (V_o_)-Zr sites to facilitate CO_2_ conversion into methanol. Additionally, the In-V_o_-Zr sites were vitally more stable toward reduction than In-V_o_-In sites in bixbyite-type In_2_O_3_, thus exhibiting superior catalytic activity and stability for CO_2_ hydrogenation to methanol. Subsequently, they further studied the nature and abundance of sites for the hydrogen dissociation on In_2_O_3_/ZrO_2_-supported catalysts (In_2_O_3_/m-ZrO_2_, In_2_O_3_/t-ZrO_2_, In_2_O_3_/am-ZrO_2_ and m-ZrO_2_:In catalysts) in CO_2_ hydrogenation to methanol [103]. The results showed that indium hydride species (In-H) and hydroxyl groups (O-H) could be found on the surface of all redox-pretreated catalysts at room temperature when they were exposed to hydrogen, and only a low concentration of hydrogen dissociation sites still existed on the surface of In_2_O_3_/m-ZrO_2_ and m-ZrO_2_:In without redox pretreatment. In_2_O_3_/m-ZrO_2(redox)_ possessed the highest concentration of surface indium sites for heterolytic activation of H_2_, and the obtained In-H species can react with CO_2_ to form surface formate species (methanol intermediates) at room temperature, indicating the appreciable reactivity of In-H and carbonates on the m-ZrO_2_ support. Additionally, the reduction in hydrogen at 400 °C led to the high dispersion of In into m-ZrO_2_ to form a m-ZrO_2_:In solid solution. Hydrogen dissociation in m-ZrO_2_:In solid solution proceeded on In^3+^-O-Zr^4+^ sites, obtaining In-H and Zr-OH species.

The preparation method of In_2_O_3_/ZrO_2_ also vitally affects the electronic structure effect, thus to optimize the interaction of In_2_O_3_ and ZrO_2_, and the surface exposure degree of In_2_O_3_, four different compositing strategies (liquid-phase coprecipitation, precipitation-coating method, ball milling method, and incipient wetness impregnation, respectively) for the synthesis of In_2_O_3_/ZrO_2_ were compared by Gao et al. [104]. It was found that the exposure area of In_2_O_3_ prepared by the precipitation-coating method was the highest (*S*_In_ = 6.22 m^2^ g^−1^), whereas it was lowest (*S*_In_ = 1.56 m^2^ g^−1^) by the coprecipitation method due to the formation of In_2_O_3_ bulk dispersion with ZrO_2_. The dispersion of In_2_O_3_ on ZrO_2_ can inhibit the over-reduction of In_2_O_3_, and the exposure area of In_2_O_3_ was beneficial for CO_2_ adsorption and activation. Furthermore, DRIFTS results and DFT calculation demonstrated that the oxygen vacancy defects of In_2_O_3_/ZrO_2_ would stabilize the key formate intermediates to facilitate the formation of methanol obeying the carbonate–formate–methoxy pathway, as shown in Figure 11: H_2_ was adsorbed on the exposed In_2_O_3_ surface (H*), and subsequently generated In-H* and O-H* by hydrogen heterolysis. CO_2_ was adsorbed and activated by In-V_o_-Zr oxygen vacancies to form carbonate species (CO_2_*), and then it combined with the activated In-H* to generate the formate intermediate (HCOO*). Later, HCOO* was further hydrogenated into CH_3_OH via the pathway of HCOO*→H_2_CO*→H_3_CO*→CH_3_OH. Apart from ZrO_2_, Ga_2_O_3_ [84], CeO_2_ [105], and MnO [106] were also used to combine with In_2_O_3_ for converting CO_2_ into methanol, and their promotion for In_2_O_3_ performance was also associated with the In_2_O_3_ dispersion, metal–support interactions, or tuning of basic sites.

## 3. Conclusions and Further Directions

In summary, In_2_O_3_-based catalysts are promising for the industrial application of thermochemical CO_2_ hydrogenation to methanol. Various research methods have been adopted to explore the formation process and possible structure of active sites and the reaction mechanism over In_2_O_3_-based catalysts for CO_2_ hydrogenation to methanol. Furthermore, research has been ongoing to further understand the structure–activity relationship and identify the key factors affecting the catalytic performance. It is commonly accepted that methanol synthesis from CO_2_ hydrogenation over In_2_O_3_-based catalysts follows a formate pathway, where CO_2_ adsorbed on the oxygen vacancy of In_2_O_3_ passes through the route of CO_2_*→HCOO*→H_2_CO*→H_3_CO*→CH_3_OH. The phase state of In_2_O_3_ plays a key role in determining CO_2_ conversion and methanol selectivity, and compared to cubic bixbyite-type In_2_O_3_ (*c*-In_2_O_3_), hexagonal In_2_O_3_ (*h*-In_2_O_3_) with a high proportion of the exposed (104) surface exhibited the higher catalytic activity and possessed high stability, which is mainly due to the facile formation of the oxygen vacancies at low coverage and the favorable formation of the hydride adsorbate at the In sites on (104) surface. The factors dictating performance improvement of In_2_O_3_-based catalysts include (1) the ability for dissociation and spillover of hydrogen, (2) the number of oxygen vacancies for CO_2_ activation, (3) the dispersion of In_2_O_3_ nanostructures, and (4) the stability of key intermediates. Two different strategies, metal doping and hybrid with other metal oxides, respectively, are utilized to optimize the above factors for enhancing the catalytic performance of In_2_O_3_-based catalysts. For the facilitation of dissociation and spillover of hydrogen, the most effective strategy is introducing the metal element (M, M = Pd, Pt, Ni or Co, etc.) into In_2_O_3_. The existence of M nanostructures sharply promotes the dissociative adsorption of hydrogen, therefore being instrumental in enhancing the hydrogenation process and increasing surface oxygen vacancy. On balance, the synergistic catalysis effect of M and In_2_O_3_ contributes to the high catalytic performance of M/In_2_O_3_ catalysts. As for the improvement of CO_2_ adsorption and key intermediates stability, supporting In_2_O_3_ on the other metal oxides is considered to be extremely useful, especially the combination of In_2_O_3_ with ZrO_2_ support. ZrO_2_ support is an excellent modifier for In_2_O_3_ to promote the concentration of oxygen vacancy, enhance the interaction with CO_2_, and stabilize the key intermediates. The structure–activity relationship of In_2_O_3_/ZrO_2_ can be concluded as follows: high surface and dispersion of In_2_O_3_ to prevent sintering and strong interaction of In_2_O_3_ and ZrO_2_ (i.e., solid solution m-ZrO_2_:In) from preventing the over-reduction of In_2_O_3_, generate more active In-oxygen vacancy (V_o_)-Zr sites for activating CO_2_ and stabilizing key formate intermediates, and also form electron-rich In_2_O_3_ (electron transfer from ZrO_2_ to In_2_O_3_) to facilitate the dissociation of hydrogen. In addition, the phase state of ZrO_2_ support greatly affects the catalytic activity of In_2_O_3_/ZrO_2_, and different from amorphous ZrO_2_ and tetragonal ZrO_2_, the interaction between monoclinic ZrO_2_ and In_2_O_3_ nanoparticles can impel atomical dispersion of In^2+^/In^3+^ into m-ZrO_2_ lattice to form solid solution m-ZrO_2_:In, which prevented the over-reduction of In species and stabilized the active In-oxygen vacancy-Zr sites to facilitate CO_2_ conversion into methanol.

Although the research of In_2_O_3_-based catalysts for CO_2_ hydrogenation to methanol has made substantial headway recently, several issues remain to be addressed in future studies. For instance, it is urgent to reveal the evolutionary process of active sites under real reaction conditions, which is extremely crucial to establish a more intuitive and reliable structure–activity relationship for designing In_2_O_3_-based catalysts. In addition, the catalytic mechanism over In_2_O_3_-based catalysts is usually proposed by theoretical study-based DFT calculation at present; however, the validity in practical applications is rather challenging. On the one hand, the microscopic reaction process (molecular level) could not be observed through experiments to verify its validity. On the other hand, the DFT calculation is unable to restore the real experimental conditions (i.e., species of active sites, mass transfer, etc.), therefore resulting in the difference between the theoretical reaction pathway and the actual reaction pathway. In order to obtain the evolutionary process of active sites and valid reaction mechanism, two considerable methods should be highlighted in future studies as follows: (1) making more efforts to analyze and identify the species of key intermediates by comprehensive in situ characterization technology (i.e., in situ DRIFTS, in situ XPS, etc.) and kinetic investigation; (2) combining DFT calculations with other simulation methods (i.e., computational fluid dynamics (CFD), kinetic Monte Carlo (KMC), etc.) to build more realistic models for theoretical study. Furthermore, from the point view of practical application, it is extremely necessary to reveal the deactivation mechanisms and enhance catalytic stability of In_2_O_3_-based catalysts in converting CO_2_ into methanol, so more attention should be paid to the issues of sintering and the structural evolution monitored by in situ/operando spectroscopic techniques. Overall, this review mainly summarized the regulation and modification of active sites of In_2_O_3_-based catalysts to facilitate the activation of reactants and stabilization of intermediates in CO_2_ hydrogenation, which is conducive to the design of more efficient In_2_O_3_-based catalysts for the highly selective transformation of CO_2_ to methanol in future studies, realizing the resource utilization of CO_2_.

## Figures and Tables

**Figure 1 materials-16-02803-f001:**
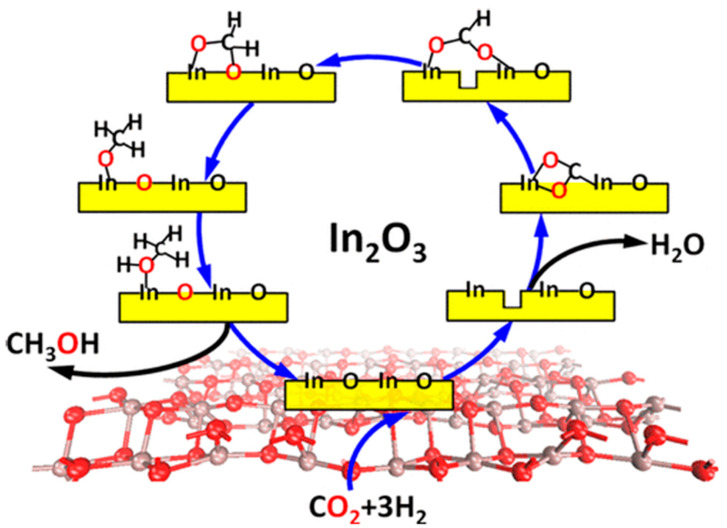
The active site and catalytic mechanism of CO_2_ hydrogenation to methanol over defective In_2_O_3_ (110). Reproduced with permission from ref. [53]. Copyright 2013 American Chemical Society.

**Figure 2 materials-16-02803-f002:**
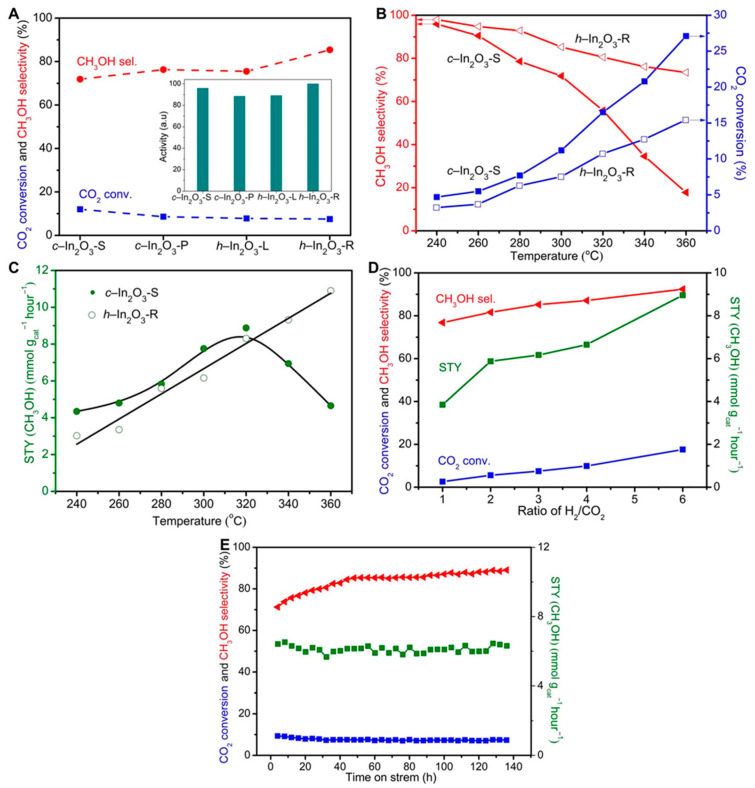
The catalytic activity of different In_2_O_3_ nanomaterials in CO_2_ hydrogenation: (**A**) CO_2_ conversion and methanol selectivity over In_2_O_3_ with different crystal phases and morphologies. (**B**) Effect of temperature on conversion of CO_2_ and selectivity of methanol over *c*-In_2_O_3_-S and *h*-In_2_O_3_-R. (**C**) Effect of temperature on space-time yield (STY) over *c*-In_2_O_3_-S and *h*-In_2_O_3_-R. (**D**) Effect of H_2_/CO_2_ molar ratio over *h*-In_2_O_3_-R. (**E**) Catalytic stability of *h*-In_2_O_3_-R. Reproduced with permission from ref. [63]. Copyright 2020 American Association for the Advancement of Science.

**Figure 3 materials-16-02803-f003:**
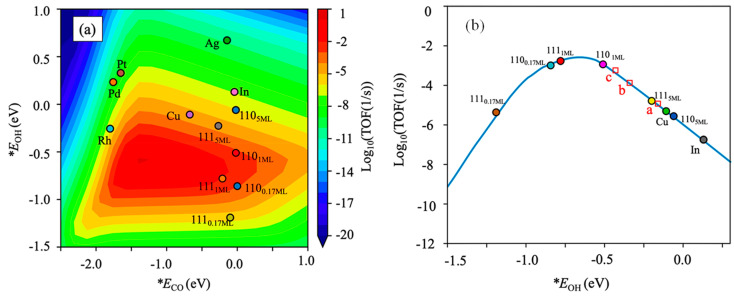
(**a**) Theoretical activity volcano of methanol synthesis from CO_2_ hydrogenation over In_2_O_3_ and transition-metal (211) surfaces. (**b**) The relationship between methanol formation and OH binding energy (fixed CO adsorption energy: −0.1 eV). Reproduced with permission from ref. [65]. Copyright 2021 American Chemical Society.

**Figure 4 materials-16-02803-f004:**
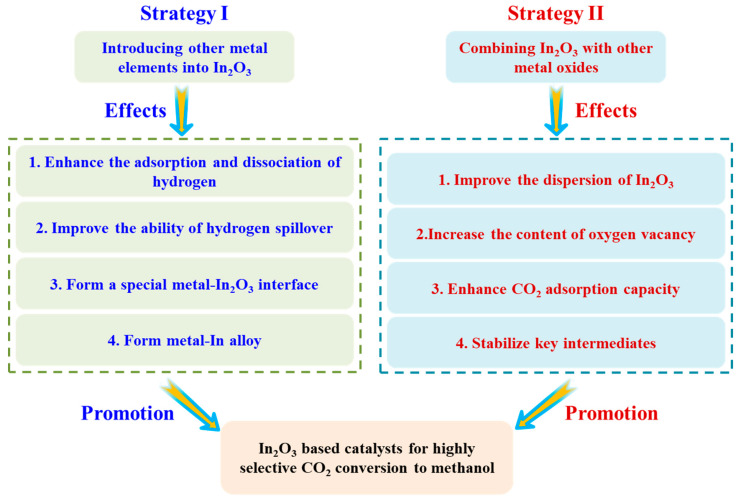
Two representative strategies for improving the performance of In_2_O_3_.

**Figure 5 materials-16-02803-f005:**
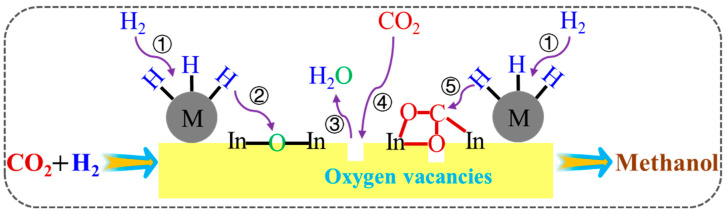
The synergistic catalysis effect of M and In_2_O_3_ in CO_2_ hydrogenation to methanol.

**Figure 6 materials-16-02803-f006:**
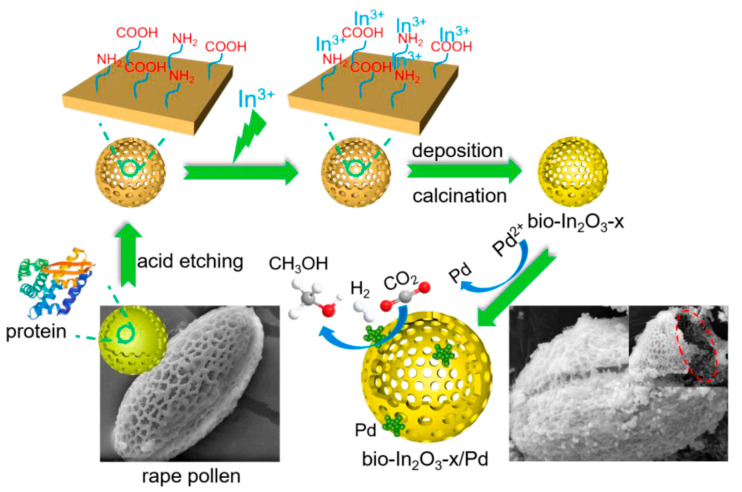
Fabrication routes of supported bio-In_2_O_3-x_/Pd catalysts. Reproduced with permission from ref. [89]. Copyright 2021 Elsevier.

**Figure 7 materials-16-02803-f007:**
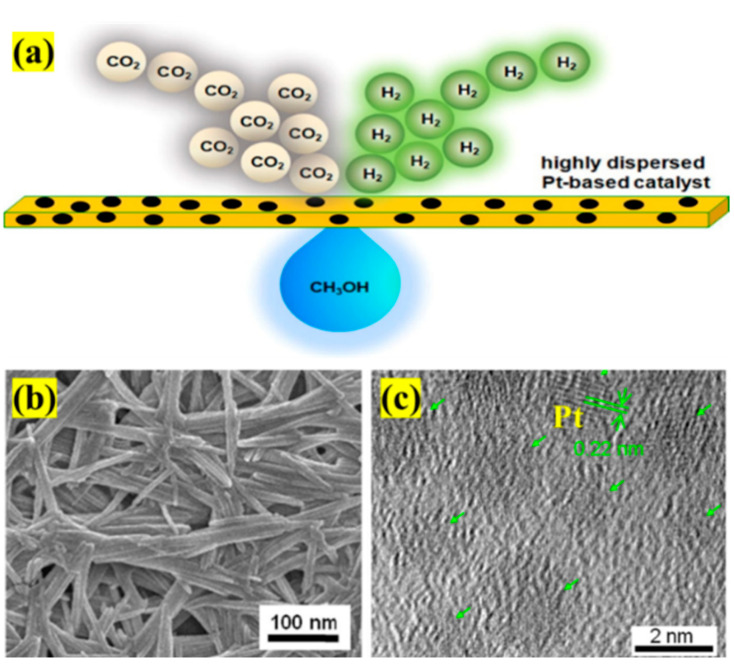
(**a**) The structure model of Pt/film/In_2_O_3_. (**b**) SEM image of Pt/film/In_2_O_3_. (**c**) HRTEM image of Pt/film/In_2_O_3_. Reproduced with permission from ref. [93]. Copyright 2019 Elsevier.

**Figure 8 materials-16-02803-f008:**
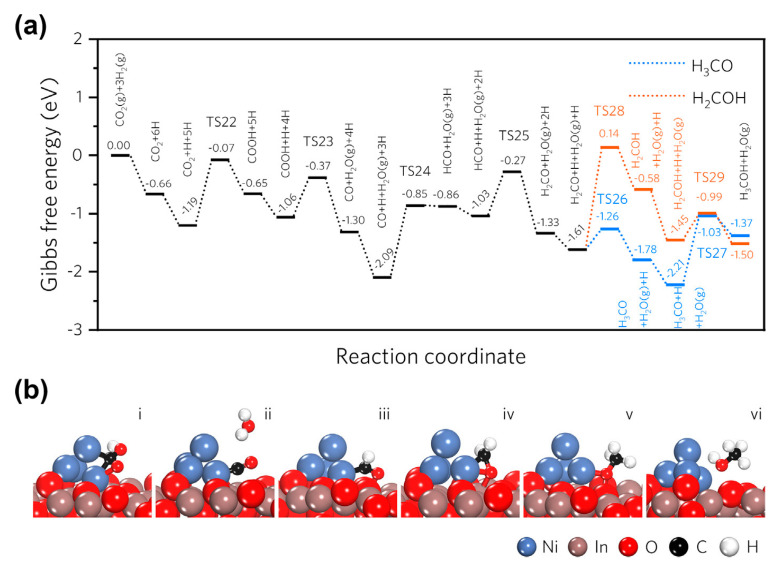
Catalytic mechanism investigation of methanol synthesis via RWGS pathway over Ni_4_/In_2_O_3_ catalyst: (**a**) calculated Gibbs free energy profile; (**b**) surface configurations of Ni_4_/In_2_O_3__D model at each elementary step. Reproduced with permission from ref. [96]. Copyright 2019 Elsevier.

**Figure 9 materials-16-02803-f009:**
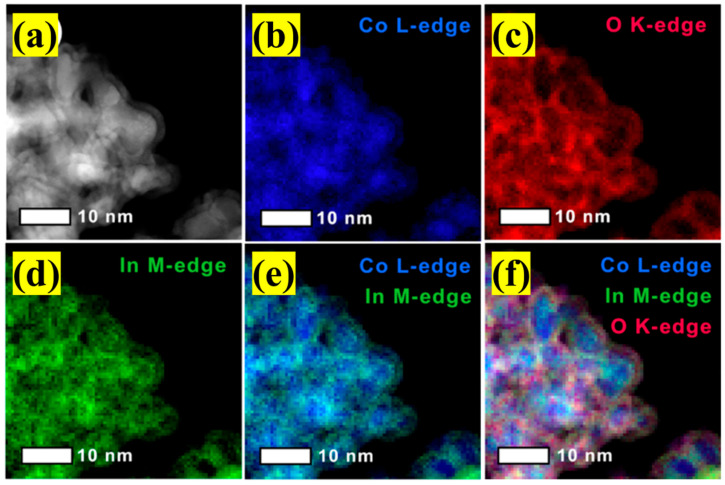
ADF-STEM imaging and elemental mapping for used 3In@8Co(300): (**a**) ADF-STEM image, (**b**) Co map, (**c**) O map, (**d**) In map, (**e**) superimposed Co/In maps, and (**f**) superimposed Co/In/O maps. Reproduced with permission from ref. [80]. Copyright 2020 American Chemical Society.

**Figure 10 materials-16-02803-f010:**
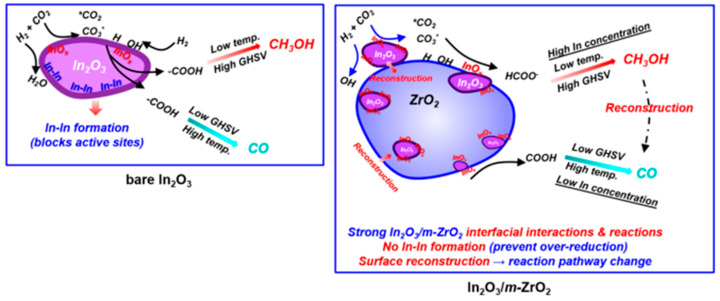
Reaction mechanism pathway on bare In_2_O_3_ and different In_2_O_3_/m-ZrO_2_ catalysts. Reproduced with permission from ref. [100]. Copyright 2022 American Chemical Society.

**Figure 11 materials-16-02803-f011:**
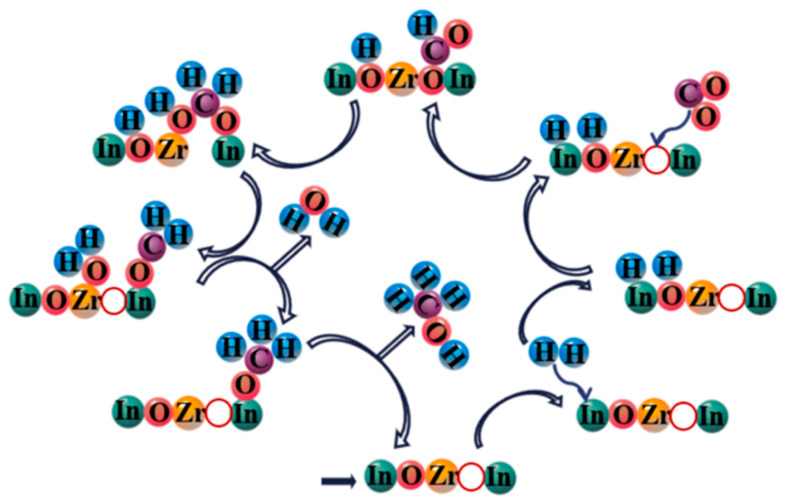
Catalytic mechanism diagram for CO_2_ hydrogenation to methanol over In_2_O_3_/ZrO_2_ catalyst prepared by precipitation-coating method. Reproduced with permission from ref. [104]. Copyright 2022 Elsevier.

**Table 1 materials-16-02803-t001:** Catalytic performance of In_2_O_3_-based catalysts in previous study.

Strategies	Catalysts	*P* (MPa)	*T* (°C)	H_2_/CO_2_ Molar Ratio	CO_2_ Conversion (%)	Methanol Selectivity (%)	STY (g_MeOH_ h^−1^ g_cat_^−1^)	Ref.
Intrinsic activity	In_2_O_3_	4	330	3:1	7.1	39.7	~0.12	[38]
	In_2_O_3_	5	300	4:1	* ^a^ *	100	~0.18	[58]
	In_2_O_3_	5	300	4:1	9.4	~62.5	~0.34	[61]
	*c*-In_2_O_3_	4	340	4:1	~12.0	~19.0	~0.09	[62]
	*rh*-In_2_O_3_	4	340	4:1	~5.0	~30.0	~0.05	[62]
	*c*-In_2_O_3_	3	300	3:1	~4.0	~70.5	0.06	[64]
	*h*-In_2_O_3_	3	300	3:1	~4.7	~71.0	0.07	[64]
	*c*/*h*-In_2_O_3_-1	3	300	3:1	~5.7	~72.3	0.09	[64]
	*c*/*h*-In_2_O_3_-2	3	300	3:1	~6.2	~73.0	0.10	[64]
	*c*/*h*-In_2_O_3_-3	3	300	3:1	~5.0	~72.1	0.08	[64]
Introducing other metal elements into In_2_O_3_	Pd-P/In_2_O_3_	5	300	4:1	~20.0	~70.0	0.89	[68]
	*h*-In_2_O_3_/Pd	3	300	3:1	~10.5	72.4	0.53	[69]
	Pd/MnO/In_2_O_3_	3	280	3:1	4.5	71.3	0.24	[70]
	Pt/In_2_O_3_	5	300	4:1	17.3	~54.0	0.54	[61]
	Pt/In_2_O_3_	4	300	3:1	5.7	~71.5	~0.75	[71]
	Rh/In_2_O_3_	5	300	4:1	17.1	56.1	0.54	[72]
	Rh-5-In_2_O_3_	5	270	4:1	10.0	71.0	0.52	[73]
	Ru/In_2_O_3_	5	300	4:1	14.3	69.7	0.57	[74]
	Au/In_2_O_3_	5	300	4:1	11.7	67.8	0.47	[75]
	Ir/In_2_O_3_	5	300	4:1	17.7	~70.0	~0.77	[76]
	Ni/In_2_O_3_	5	300	4:1	18.4	~54.0	0.55	[77]
	Ni/In_2_O_3_	3	250	3:1	3.0	~52.0	~0.25	[78]
	Co/In_2_O_3_	4	300	3:1	~9.0	~40.0	~0.31	[79]
	In_2_O_3_@Co_3_O_4_	5	250	4:1	8.3	~87.0	0.65	[80]
	Cu_11_In_9_-In_2_O_3_	3	260	3:1	10.3	86.2	~0.19	[81]
Combining In_2_O_3_ with other metal oxides	In_2_O_3_/ZrO_2_	5	300	4:1	* ^a^ *	~100	~0.31	[58]
	In_2_O_3_/m-ZrO_2_	3	280	3:1	12.1	84.6	* ^a^ *	[82]
	In_2.5_/ZrO_2_	5	280	4:1	* ^a^ *	60.0	~0.07	[83]
	Ga_0.4_In_1.6_O_3_	3	320	3:1	~12.0	~28.0	* ^a^ *	[84]
	InCe oxides	0.1	290	3:1	* ^a^ *	~10.0	~0.12 *^b^*	[85]
	In_2_O_3_/Al_2_O_3_	5	280	4:1	* ^a^ *	* ^a^ *	~0.04	[86]

*^a^* Not available. *^b^* μmol_MeOH_ s^−1^ g_In_^−1^.

## Data Availability

Not applicable.

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
