# Peer review of "Recent Advances of Indium Oxide-Based Catalysts for CO2 Hydrogenation to Methanol: Experimental and Theoretical"

_materials, 2023, doi:10.3390/ma16072803_

Round 1

Reviewer 1 Report

The paper appears both well structured and written. It highlights the limitation shown by the use  of indium oxide in the hydrogenation reaction of CO2 to methanol ( usually high selectivity but low CO2 conversion) at the same time evaluates the possible strategies to overcome this limitation through the incorporation of promoters. The consideration on reaction mechanism are considered interesting.

Author Response

Respond to Reviewer #1: The paper appears both well structured and written. It highlights the limitation shown by the use of indium oxide in the hydrogenation reaction of CO2 to methanol (usually high selectivity but low CO2 conversion) at the same time evaluates the possible strategies to overcome this limitation through the incorporation of promoters. The consideration on reaction mechanism are considered interesting.

R: Thank you for your review. 

Reviewer 2 Report

The review article presents recent advances of indium oxide-based catalysts for CO2 hydrogenation to methanol in the case experimental and theoretical. The paper is excellent, well-written and offers valuable insights that could benefit the scientific community. The methodology employed is rigorous, and the article adheres to the journal's standards, though a few minor grammatical errors in sentence structure require correction. However, the paper's literature review appears insufficient. Therefore, I recommend a minor revision before considering it for publication.

Author Response

Respond to Reviewer #2: The review article presents recent advances of indium oxide-based catalysts for CO2 hydrogenation to methanol in the case experimental and theoretical. The paper is excellent, well-written and offers valuable insights that could benefit the scientific community. The methodology employed is rigorous, and the article adheres to the journal's standards, though a few minor grammatical errors in sentence structure require correction. However, the paper's literature review appears insufficient. Therefore, I recommend a minor revision before considering it for publication.

R: Thank you for your valuable suggestion. We have added some references in the revised manuscript.

Reviewer 3 Report

I have carefully read this paper entitled “Recent advances of indium oxide-based catalysts for CO2 hydrogenation to methanol: Experimental and theoretical". As a result, I have only a few minor points that the authors should address before it is accepted for publication. Please, publish subject to the following revisions:

1-      Rewrite the novelty statement and importance of this work at the end of the introduction section.

2-      The abstract and conclusion should be rewritten and showed more clear results and the novelty of this study.

3-      Please provide more explanation and discussion about why Noble metal/In2O3 catalysts could improve the CO2 hydrogenation to methanol

Author Response

Respond to Reviewer #3: I have carefully read this paper entitled “Recent advances of indium oxide-based catalysts for CO2 hydrogenation to methanol: Experimental and theoretical". As a result, I have only a few minor points that the authors should address before it is accepted for publication. Please, publish subject to the following revisions:

1.Rewrite the novelty statement and importance of this work at the end of the introduction section.

R: Thank you for your valuable suggestion. We have rewritten the novelty statement and importance of this work at the end of the introduction section (lines 85-98).

2.The abstract and conclusion should be rewritten and showed more clear results and the novelty of this study.

R: Thank you for your valuable suggestion. We have modified the abstract (lines 14-27, and 29-31) and conclusion (lines 582-587, 597-598, 609-614, and 638-643) according to your suggestion.

3.Please provide more explanation and discussion about why Noble metal/In2O3 catalysts could improve the CO2 hydrogenation to methanol

R: Thank you for your valuable suggestion. We have provided the explanation and discussion about why Noble metal/In2O3 catalysts could improve the CO2 hydrogenation to methanol (lines 231-240).  
